# Automatic Medical Face Mask Detection Based on Cross-Stage Partial Network to Combat COVID-19

**Christine Dewi [1],\* and Rung-Ching Chen [2],\***

[1]  Department of Information Technology, Satya Wacana Christian University, Salatiga 50711, Indonesia
[2]  Department of Information Management, Chaoyang University of Technology, Taichung 41349, Taiwan
\*  Correspondence: christine.dewi@uksw.edu (C.D.); crching@cyut.edu.tw (R.-C.C.)

**Abstract:** According to the World Health Organization (WHO), the COVID-19 coronavirus pandemic has resulted in a worldwide public health crisis. One effective method of protection is to use a mask in public places. Recent advances in object detection, which are based on deep learning models, have yielded promising results in terms of finding objects in images. Annotating and finding medical face mask objects in real-life images is the aim of this paper. While in public places, people can be protected from the transmission of COVID-19 between themselves by wearing medical masks made of medical materials. Our works employ Yolo V4 CSP SPP to identify the medical mask. Our experiment combined the Face Mask Dataset (FMD) and Medical Mask Dataset (MMD) into one dataset to investigate through this study. The proposed model improves the detection performance of the previous research study with FMD and MMD datasets from 81% to 99.26%. We have shown that our proposed Yolo V4 CSP SPP model scheme is an accurate mechanism for identifying medically masked faces. Each algorithm conducts a comprehensive analysis of, and provides a detailed description of, the benefits that come with using Cross Stage Partial (CSP) and Spatial Pyramid Pooling (SPP). Furthermore, after the study, a comparison between the findings and those of similar works has been provided. In terms of accuracy and precision, the suggested detector surpassed earlier works.

**Keywords:** object recognition; Convolutional Neural Network (CNN); COVID-19; medical face mask; Yolo; deep learning

## 1. Introduction

The coronavirus COVID-19 wreaked havoc on humanity last year regardless of age, gender, or geographic location. For a brief period, the virus brought the entire planet to a halt. In addition to causing physical hardship, COVID-19 also contributed to economic crises in many developed and developing countries as well as in many third-world countries [1,2]. As a result of the epidemic of COVID-19, numerous nations have introduced new restrictions regarding the usage of face masks as a method of infection prevention. In the years leading up to COVID-19, people developed the habit of wearing masks to protect themselves from the harmful effects of air pollution; this practice has continued into the present day [3].

When others are self-conscious about their appearance, they hide their feelings from the public by covering their faces. In addition, when dealing with patients suffering from respiratory infections, medical professionals often use face masks as part of their droplet prevention measures. The rational use of surgical face masks, when exposed to high-risk areas, would be a reasonable recommendation for those who are particularly vulnerable.

Because evidence suggests that COVID-19 can be transmitted before symptoms appear, wearing face masks by everyone in the community, including those who have been infected but are asymptomatic and contagious, may help to reduce the spread of the

disease [4,5]. The rapid spread of COVID-19 led the World Health Organization (WHO) to declare COVID-19 a worldwide pandemic by 2020 because of the virus's rapid transmission. The surgical mask has some benefits, as follow: (1) To prevent the spread of respiratory viruses from sick people to healthy people, surgical masks are often worn during surgery. A surgical mask should be always worn by anyone with asthma symptoms when exposed to the elements even if they have only mild symptoms. (2) It is recommended that one wear a surgical mask whenever they travel by public transport or live in overcrowded areas. To ensure proper mask wear and removal, it is critical to practice good hand hygiene both before putting on and after removing the mask. Since many nations have laws forcing people to wear face masks in public areas, masked face identification is vital for face applications such as object detection and surveillance [6,7].

In addition, there is an immediate need for research into the length of time that face masks can provide protection [8] as well as efforts to lengthen the time that disposable masks can be used, and the creation of reusable masks should be promoted. As Taiwan can stockpile large quantities of face masks, other countries or regions may now consider using this capability as part of their pandemic preparedness measures should new pandemic occur in the future. According to the World Health Organization (WHO), for the government to successfully fight and win the battle against the COVID-19 pandemic, the government must provide instructions and supervision to the public in public places, especially in densely populated areas. Ensuring that face mask laws are complied with is part of this. As an example, the integration of surveillance systems with artificial intelligence models could be used in this situation [9].

Deep transfer learning and the combination of CSP and SPP [10] were used in this article to develop a mask face identification model. Considering the ability of the proposed model to distinguish people who are not wearing masks, this model might be integrated with security cameras to stop the transmission of COVID-19. Researchers are primarily interested in medically disguised faces to prevent the spread and transmission of the coronavirus, particularly COVID-19. With the rapid growth of deep learning, several object detectors have recently been developed. A unique object detection algorithm, Yolo V4, has been proposed by Wang et al. [11]. In terms of both accuracy and processing speed, the Yolo V4 object detection approach exceeds the conventional method that is currently in use for object detection.

The following are the most significant contributions made by the paper.

(1)  An innovative deep learning detector model that automatically identifies and localizes a medically masked face on an image has been developed and demonstrated.
(2)  Identification and evaluation of the advantages and disadvantages of using the Yolo V3, Yolo V4, and Yolo V5 facial recognition systems for the detection and recognition of medical face masks.
(3)  Our work combined the Cross Stage Partial network (CSP) and Spatial Pyramid Pooling (SPP) with the Yolo model.
(4)  This work performs a comparative analysis of the combination of the Yolo V3, Yolo V4, and Yolo V5 models.

The following is an outline of the paper. Section 2 describes and examines previous related works. Section 3 provides a brief description of our proposed methodology. Section 4 presents the dataset, training data, and system test results. Section 5 concludes with recommendations for further research and development.

## 2. Related Works

### 2.1. Medical Face Mask Detection with Deep Learning

In general, while people are wearing face masks, high attention is paid to the construction of their faces and the recognition of their identities. In [12], the researchers were interested in identifying those who are not using face masks to aid in the prevention and decrease of the transmission and spread of the COVID-19 virus and other diseases.

Principal component analysis (PCA) is presented on masked and non-masked face recognition datasets, and a comparison of the two methods is presented in [13]. As part of their research, they have identified statistical strategies that can be used in maskless face identification and masked face recognition techniques. PCA is a statistical technique that is more effective and successful than others and is commonly used. The authors of [14] are concentrating on the unmasking of a masked face, which is a novel thought with significant practical implications. A GAN-based network [15] with two discriminators was used in their research, in which one discriminator assisted in learning the general structure of the face and another discriminator was introduced to focus learning on the deep missing region.

In other research, LLE-CNNs for masked face detection are presented. These LLE-CNNs have three primary modules that make them up. The proposed module begins by combining two pre-trained CNNs to identify potential face regions from the input image and describe them with greater descriptors, which are then used to refine the suggestion. A consistency descriptor is created by applying the locally linear embedding (LLE) technique and dictionaries that have been learned on a huge pool of generated ordinary faces, masked faces, and non-faces among other methods, in the Embedding module [16]. In [17] demonstrate a hybrid face mask detection model that incorporates both deep and traditional machine learning techniques. The recommended model is divided into two sections. It is meant for usage in conjunction with the Resnet50 feature extraction technique, which is the initial component of this setup. Although both components are employed in the classification of face masks, the second component is specifically meant for the classification phase utilizing decision trees, support vector machines (SVM), and an ensemble approach.

In [18], the authors used the Yolo V3 algorithm for face detection. Furthermore, Yolo V3 is built on Darknet-53, which serves as its backbone. The proposed approach had an accuracy of 93.9% in testing. The authors in [18] discussed the development of a system for detecting the presence or absence of a mandatory medical mask in the operating room. The primary goal is to minimize the number of false positive face detections as much as is reasonably possible while also ensuring that mask detections are not missed. This will ensure that alerts are only triggered for medical workers who are not wearing a surgical mask while performing their duties. The suggested system was archived with 95% precision.

### 2.2. Yolo Algorithm

In 2016, Redmon and Farhadi made a proposal for the Yolo V3 [19]. It divides the input image into ($N \times N$) grid cells [20] of the same size and forecasts bounding boxes and probabilities for each grid cell. When it comes to developing predictions, Yolo V3 takes advantage of multi-scale integration, and a single neural network is employed to generate the whole overview that is provided. Yolo V3 allows for the creation of a unique bounding box anchor for every ground truth item [21]. Furthermore, Yolo V4 was released by [22] in 2020. The Yolo V4 structure is as follows: (1) Backbone: CSPDarknet53 [23]. (2) Neck: SPP [24], Path Aggregation Network (PAN) [25]. (3) Head: Yolo V3 [19]. In the backbone, Yolo V4 utilizes a Mish [26] activation function.

Yolo technique [27] is a common end-to-end system creation approach that is used in many applications. This algorithm is more compact than the R-CNN algorithm in terms of time [28,29]. Yolo V4 is an implementation of dense prediction in the head that only requires one stage and makes use of the Yolo V3 algorithm. Yolo V3 also divides the input image into $m \times n$ grids cells of the same size, which are then combined to form a final image [20,30]. The Yolo V5 release represents a significant departure from earlier versions. PyTorch is being used as an alternative to Darknet at this time. This typically relies on CSPDarknet53 as its primary support mechanism in order to carry out its operations. It solves the problem of repeated gradient information when it is used in big backbones and integrates gradient change into the feature map. This results in an increase in inference

speed while simultaneously increasing accuracy and shrinking the model size by using the fewest possible parameters. Specifically, it uses the route aggregation network (PANet) to act as a chokepoint and speed up data transfer. To achieve its aims of improved performance and reliability, PANet employs a novel feature pyramid network (FPN) with a large number of bottom-up and top-down layers as well as a variety of top-down levels. As a result, the model becomes better at conveying details at a lower level. Improved accuracy of object localization in lower layers is a result of the use of PANet, which also enhances the accuracy of object localization in higher levels.

Aside from this, the head in Yolo V5 is identical to the heads in Yolo V4 and Yolo V3, and it generates three different outputs of feature maps to achieve multiscale prediction [31,32]. The Yolo V5 approach is superior to the Yolo V1 algorithm since it does not necessitate the utilization of all of Yolo V1's connection layers. The darknet-19 network model is a C++ implementation for extracting the depth characteristic of the target picture [33].

*2.3. Cross Stage Partial (CSP) Networks and Spatial Pyramid Pooling (SPP)*

A novel strategy for locating objects has recently been proposed in the form of Yolo V4, an object detection system that is founded on CSP [34]. This research describes a method for scaling networks that alters not just their dimensions but also their internal architecture as well as their width, depth, and resolution. As a result of this research, Scaled-Yolo V4 was built. Yolo V4 is a real-time object detection system that runs on a general-purpose graphics processing unit (GPU). Jiang et al. [34] redesigned Yolo V4 into Yolo V4-CSP to obtain the best speed–accuracy trade-off. Enhancing CNN's learning capabilities while decreasing the computing bottleneck and memory cost is achieved with the help of CSP [11], which is the Yolo-V4 backbone network. Because of its portability and light weight, it can be used in any setting. This paper made use of CSP blocks and SPP model in order to apply to the DarkNet53 network with Yolo V3 and Yolo V4.

SPP [24] has the following advantages: Regardless of the input dimensions, SPP can produce an appropriate fixed-length output [35,36]. Another key difference between SPP and sliding window pooling is that SPP uses many window sizes (spatial bins), whereas the latter only uses one. An SPP block layer was included in the configuration files of Yolo V3, Yolo V4, and Yolo V5 in order to facilitate the conduct of this experiment. We can also create spatial models with them by using the same SPP block layers that are in the configuration file. The spatial model takes advantage of down-sampling in the convolutional layers in order to obtain the necessary properties in the max-pooling layers, which are then used to construct the model. This is done to save time and improve accuracy. This applies three different sizes of the max pool for each image by using [route]. Different layers, -2; -4; and -1, -3, -5, -6 in $conv_5$, were used in each [route]. Furthermore, Figure 1 illustrates the CSP and SPP architecture.

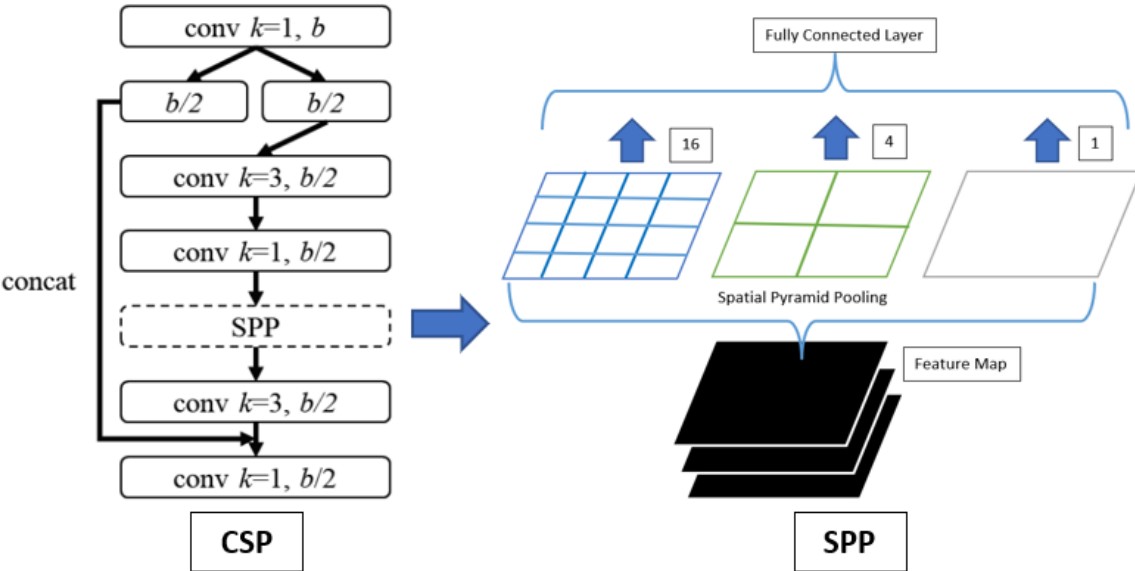

**Figure 1.** CSP and SPP architecture.

## 3. Methodology

### 3.1. Yolo V4 CSP SPP

The proposed methodology is to recognize face medical masks based on Yolo V4. Figure 2 describes the Yolo V4 CSP SPP architecture. The process of recognizing a medical mask on a person's face with Yolo V4 works as follows.

- Organizes the input image into $m \times m$ grids, with each grid generating $K$ bounding boxes based on the calculation of the anchor boxes in the previous grid.
- Makes use of the CNN to collect all of the object characteristics from the picture and predict the $b = [b_x, b_y, b_w, b_h, b_c]^T$ and the $class = [class\ M1_1, class\ M2_2, \ldots., class\ MC_c]^T$. Given the anchor box of size $(p_x, p_y)$ at the grid cell with its top left corner at $(c_x, c_y)$, the model predicts the offset and the scale $(t_x, t_y, t_w, t_h)$, and the corresponding predicted bounding box $b$ has center $(b_x, b_y)$ and size $(b_w, b_h)$. The confidence score is the sigmoid (σ) of another output $t_o$.
- Compares the maximum confidence $IoU_{pred}^{truth}$ of the $K$ bounding boxes with the threshold $IoU_{thres}$.
- If $IoU_{pred}^{truth} > IoU_{thres}$, this means that the object is contained in the bounding box. If this is not the case, the item is not in the bounding box.
- The object category should be chosen based on the category with the highest anticipated probability.
- The Non-Maximum Suppression (NMS) method is then used to perform an optimum search strategy to suppress duplicate boxes and outcomes, after which the outcomes of object recognition are displayed on the screen.

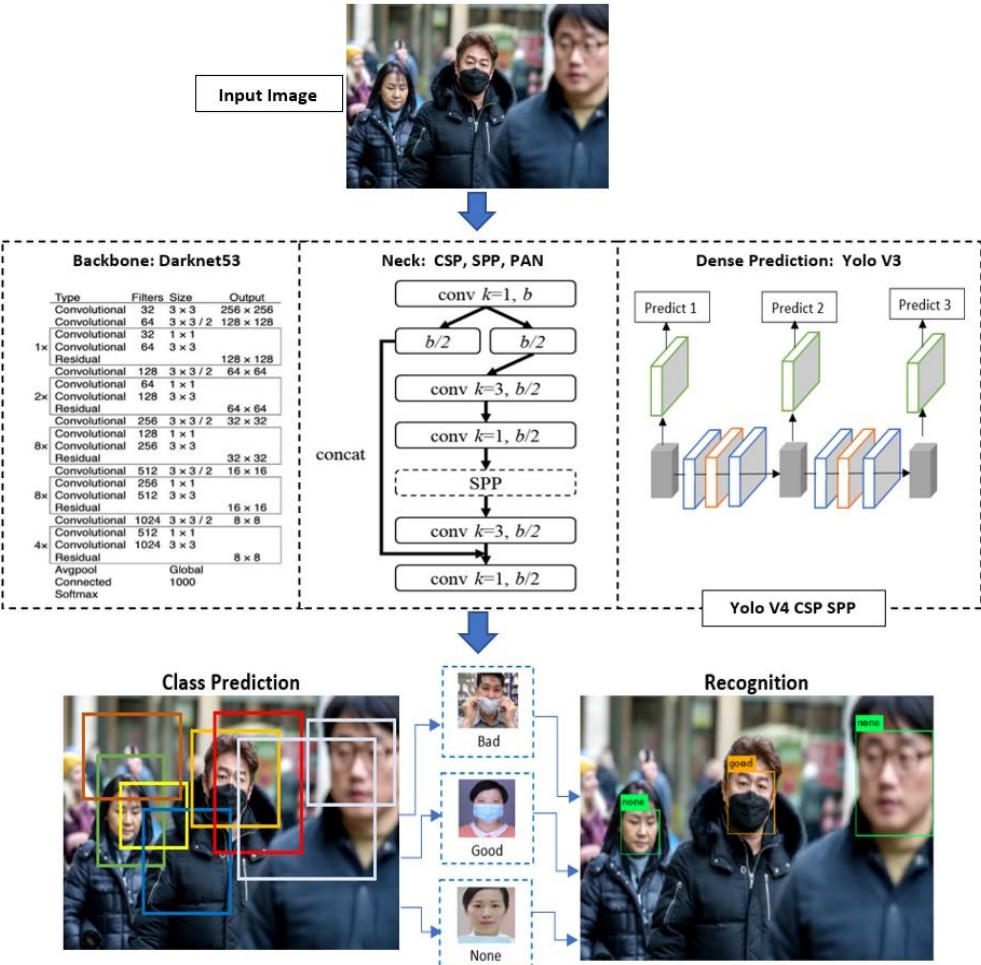

**Figure 2.** System Architecture of Yolo V4 CSP SPP. Source of the human faces: The FMD and MMD datasets.

Yolo V4 with a cross-stage partial network and spatial model is used in this work to obtain the key information in the max-pooling layers by down sampling in the convolutional layers. Yolo V4 CSP SPP contains of the following sections: (1). Darknet53 as a backbone. (2). CSP, SPP, PAN in the neck. (3). Yolo V3 as a dense prediction. The Yolo V3, Yolo V3 SPP, Yolo V3 CSP SPP, Yolo V4, Yolo V4 CSP SPP, and Yolo V5 frameworks all perform face medical mask detection and recognition in a single step. Bounding boxes are the most often encountered type of annotation in deep learning, and they outnumber all other types. Bounding boxes are rectangular boxes that are used in computer vision to describe the location of the object that is being examined. These coordinates, which are located at the upper-left corner of the rectangle as well as the lower-right corner of the rectangle, can be used to determine their *x* and *y* axis coordinates. Bounding boxes are frequently utilized in the context of object detection and localization tasks. To generate a bounding box for each sign, the BBox label tool [37] is employed.

Three distinct types of labels are applied throughout the labeling procedure (0, 1, 2). In contrast to other input formats, Yolo input values are not represented by object coordinates. However, the Yolo input data is the position of the object's center point, as well as its width and height (*x, y, w, h*). Bounding boxes are typically expressed by two coordinates—e.g., (*x1, y1*) and (*x2, y2*)—or by one coordinate—e.g., (*x1, y1*)—and the width (*w*) and height (*h*) of the bounding box. Equations (1)–(6) show the transformation process.

$$dw = 1/W \tag{1}$$

$$x = \frac{x1 + x2}{2} \times dw \tag{2}$$

$$dh = 1/H \tag{3}$$

$$y = \frac{y1 + y2}{2} \times dh \tag{4}$$

$$w = (x2 - x1) \times dw \tag{5}$$

$$h = (y2 - y1) \times dh \tag{6}$$

where $W$ is the width of the image and $H$ is the height of the image. For each image file in the same directory, an *a.txt* file with the same name will be created. Each *a.txt* file contains the object class, object coordinates, image file height and width, and other metadata.

### 3.2. FMD and MMD Dataset

The experiments in this paper were conducted using two publicly available medical face mask datasets. First, the Face Mask Dataset (FMD) in [38] is the publicly available masked face dataset. This dataset contains 853 images that make up the FMD dataset and all in the PASCAL VOC format. Figure 3a depicts some FMD sample images. Next, the Medical Masks Dataset (MMD) is available on Kaggle [39]. The MMD dataset consists of 682 images, each of which contains over 3000 medically masked faces. Figure 3b shows examples of images in MMD, and in this experiment, all informed consent was obtained.

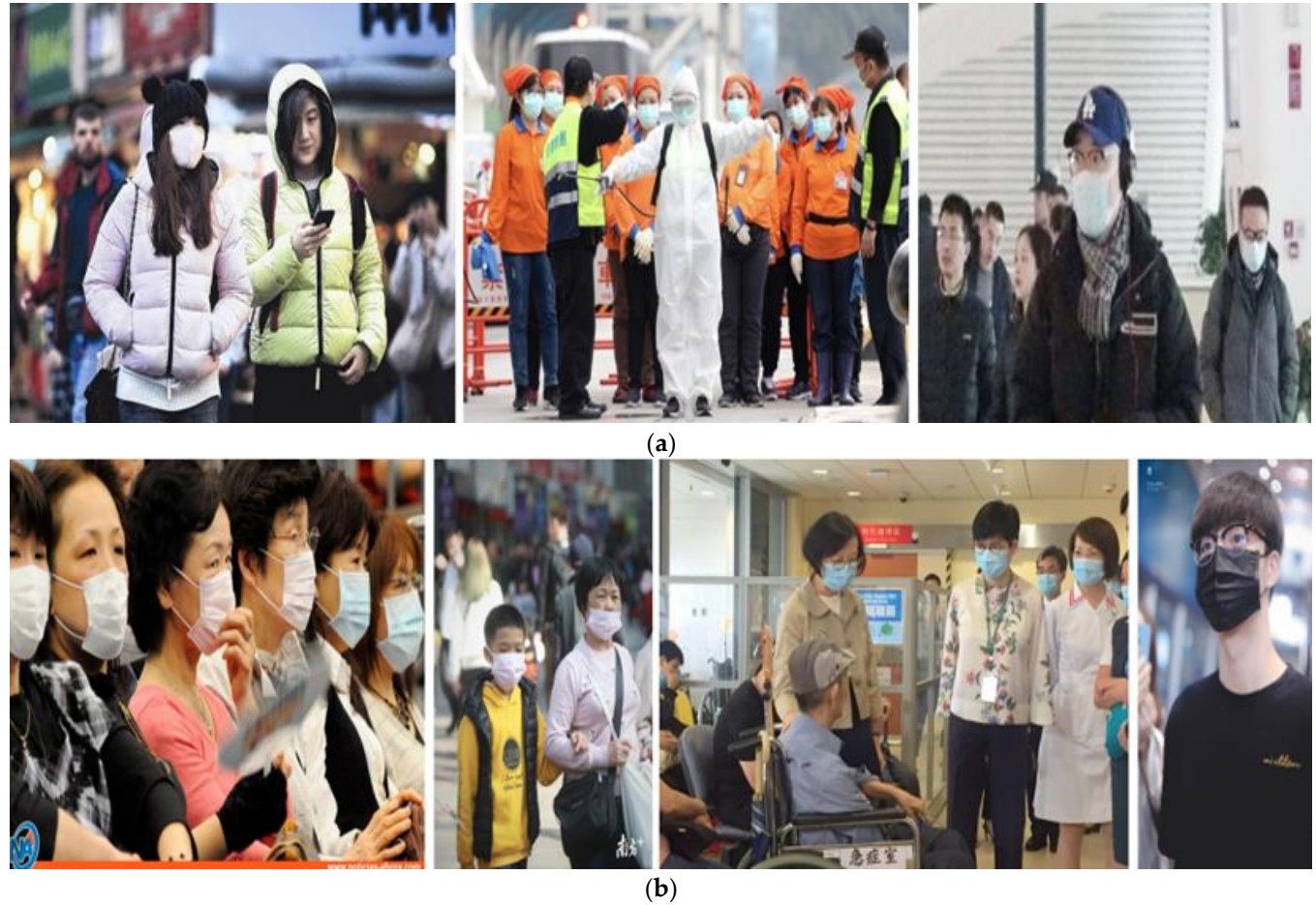

**Figure 3.** Sample of dataset in the experiment. (**a**) Face Mask Dataset (FMD) and (**b**) Medical Mask Dataset (MMD).

MMD and FMD were combined in this experiment to provide a unique dataset. In total, 1415 photos were combined from the given dataset, which was accomplished by deleting low-quality images and duplicates from the source dataset. Figure 4 illustrates the combination of MMD and FMD in our works. The MMD dataset consists of 3 classes, with class names bad, good, and none. On the other hand, the FMD dataset contains 3 classes with the names *mask_wear_incorrect*, *with_mask*, and *without_mask*. Our experiment explains the 3 class as follows: *bad = mask_wear_incorrect*, *good = with_mask*, and *none = without_mask.* Figure 5 depicts the labels of the MMD and FMD datasets, which contain 3 classes, namely bad, good, and none. The *'bad'* class consists of almost 500 instances, the *'good'* class has more than 4000 instances, and the *'none'* class has approximately 500 instances. The *x* and *y* values range from 0.0 to 1.0, while the width is from 0.0 to 0.6, and the height is from 0.0 to 0.8. Masks are crucial in protecting people's health against respiratory infections because they are one of the only prophylactic methods available for COVID-19 in the absence of immunization. With this dataset, it is possible to develop a model that can distinguish between those who are wearing masks and those who are not or who are wearing masks incorrectly. In the Yolo format, each JPEG image file is accompanied with a text file with the same name but with an *a.txt* extension. This text file includes information about the location of each item in the image, including its class, *x*, *y* coordinates, width, and height.

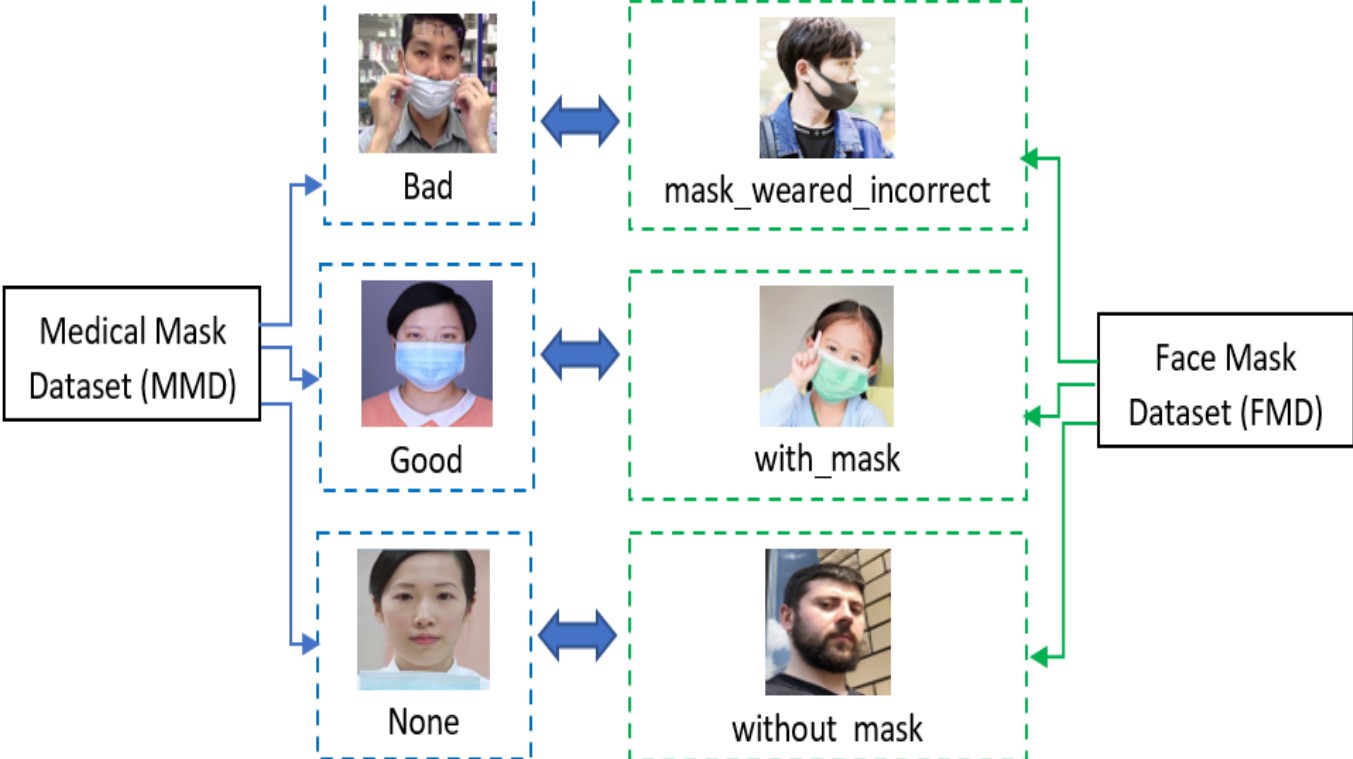

**Figure 4.** The combination of MMD and FMD datasets. Source of the human faces: The FMD and MMD datasets.

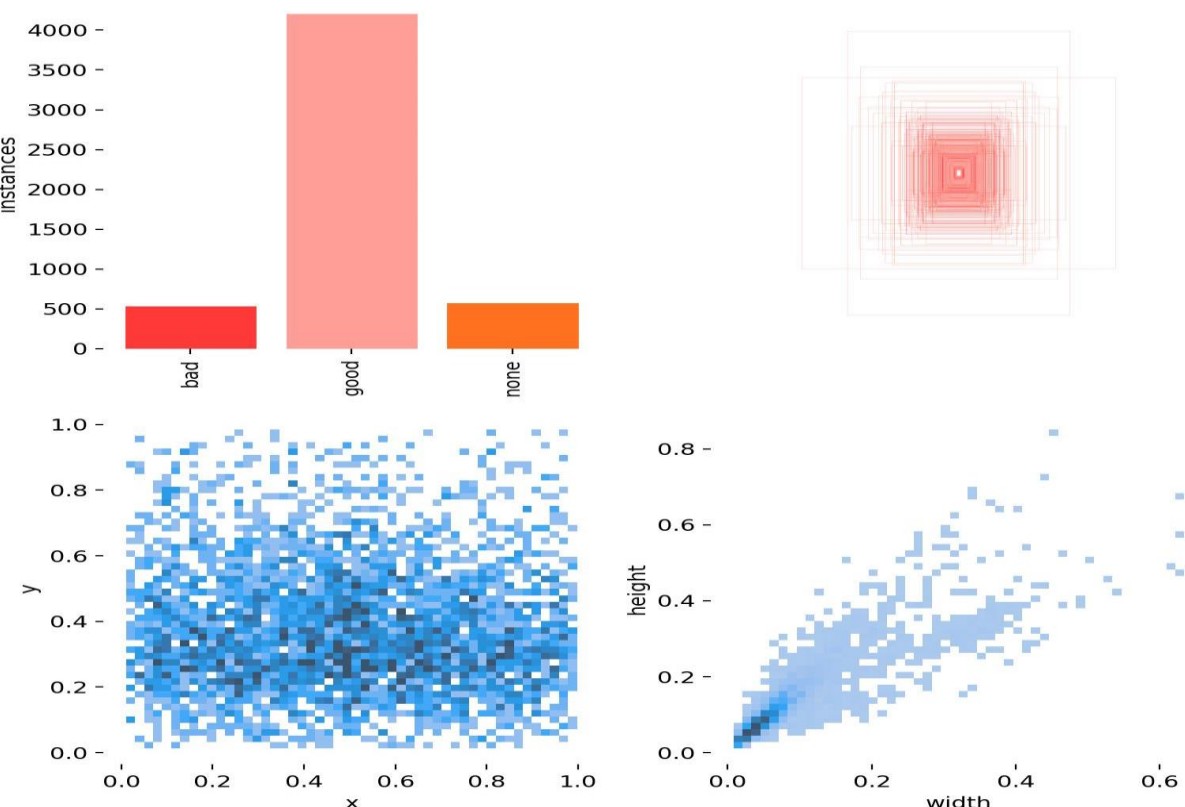

**Figure 5.** Labels of MMD and FMD dataset.

### 3.3. Training Result

Throughout the training procedure, we apply data augmentation strategies, such as padding, cropping, and horizontal flipping. As a result of their useful features, these techniques are frequently employed in the development of massive neural networks. In addition, the training model environment is an Nvidia RTX3080Ti GPU accelerator 11 GB memory, i7 Central Processing Unit (CPU), and 16 GBDDR2 memory. Real-time detection is a top objective for Yolo V4 CSP SPP, which conducts training on a single graphics processing unit (GPU).

Observations outside of the training set are needed to detect a machine learning model's behavior. This would prevent the evaluation of the model from being biased. Using training observations for model evaluation is similar to giving a class a set of questions and then using several of those questions on a final exam. There is no way to know if students understand the material or just memorize it. Splitting the data into two distinct groups is the quickest and easiest approach. Then, we put one to use in model training and the other in testing. The term for this approach is "holdout". In this experiment we set seventy percent of the dataset is used for training purposes, while the other thirty percent is used for testing.

Figure 6 depicts the training procedure for the Yolo V4 CSP SPP model. When training and detecting, all images will be scaled down to the network size width = 416 and height = 416, as provided in the training configuration file. The training will be processed for 9000 batches, with the max batches set to 9000, the policy set to steps, and the steps set to 7200 and 8100. The learning speed is determined by a hyperparameter that dictates how much the model must change in response to the predicted error. This parameter is updated with each update of the model weights. Choosing a learning level can be difficult because too low of a value can result in a lengthy training process that can become bogged down, but too large of a value can result in learning a suboptimal weight series too quickly or training unstable phases that can lead to failure. The learning rate of a neural network

model determines how quickly or slowly it learns to solve a problem. The learning rate will begin at 0.001 and will be multiplied by scales to obtain the new learning rate after that. Yolo V4 comes to a halt at an average loss of 0.5766 during the training stage of the medical mask and face mask datasets.

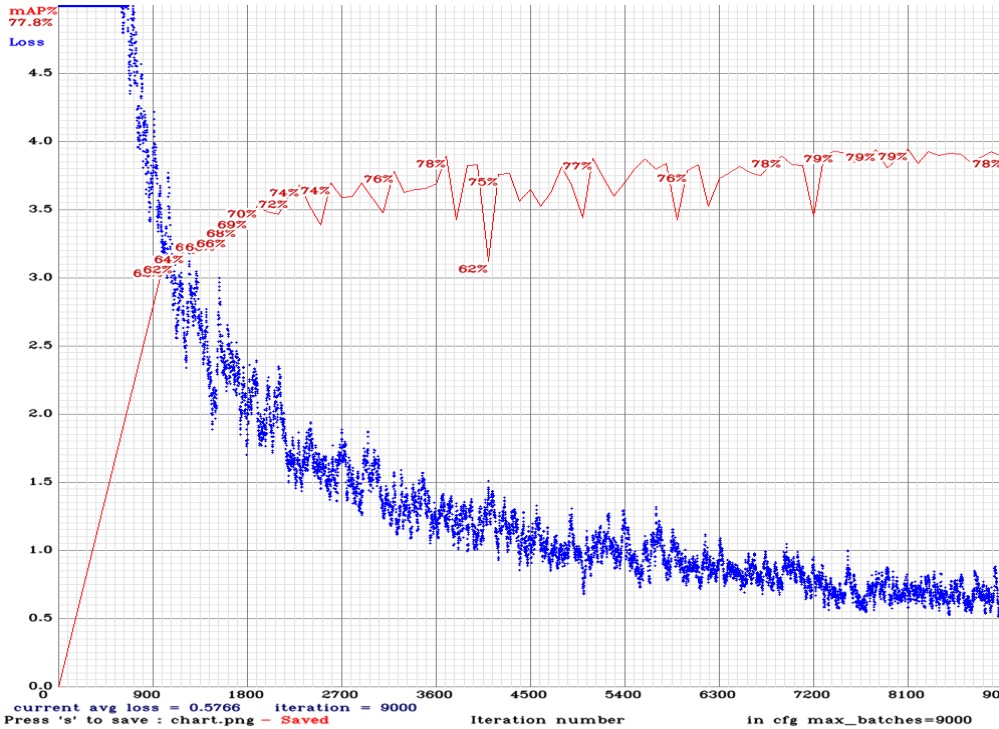

**Figure 6.** Training performance using Yolo V4 CSP SPP.

All classes' post-training performances are shown in Table 1, which includes the training loss value, *mAP, AP*, precision, recall, F1, and *IoU* performance for each class. Yolo V4 CSP SPP obtains the maximum average *mAP* of 78.48% and *IoU* 75.9%, followed by Yolo V4, with *mAP* 67.9%; Yolo V5, with *mAP* 63.5%; Yolo V3 CSP SPP, with *mAP* 59.91%; Yolo V3 SPP, with *mAP* 59.76%; and Yolo V3, with *mAP* 58.86%.

**Table 1.** Training performance for all models with FMD and MMD datasets.

| Model | Class ID | *AP* | Precision | Recall | F1-Score | *IoU* (%) | *mAP*@0.50 (%) |
|---|---|---|---|---|---|---|---|
| Yolo V3 | 0 | 52.18 | 0.82 | 0.78 | 0.8 | 64.83 | 58.86 |
| | 1 | 89.33 | | | | | |
| | 2 | 35.09 | | | | | |
| Yolo V3 CSP SPP | 0 | 56.4 | 0.78 | 0.81 | 0.79 | 60.25 | 59.91 |
| | 1 | 90.36 | | | | | |
| | 2 | 32.97 | | | | | |
| Yolo V3 SPP | 0 | 59.05 | 0.81 | 0.79 | 0.8 | 61.57 | 59.76 |
| | 1 | 91.09 | | | | | |
| | 2 | 29.16 | | | | | |
| Yolo V4 | 0 | 65.67 | 0.78 | 0.93 | 0.85 | 66.8 | 67.9 |
| | 1 | 96.47 | | | | | |
| | 2 | 41.56 | | | | | |
| Yolo V4 CSP SPP | 0 | 76.88 | 0.89 | 0.91 | 0.9 | 75.9 | 78.84 |
| | 1 | 96.06 | | | | | |
| | 2 | 63.57 | | | | | |
| Yolo V5 | 0 | 47.8 | 0.62 | 0.77 | 0.687 | 52.3 | 63.5 |

| 1 | 94.6 |
|---|------|
| 2 | 48.1 |

Based on Table 1, it is possible to draw the conclusion that the CSP and SPP layers have the potential to improve the performance of Yolo V3 and YoloV4 training. During the training phase, our model can make use of CSP and SPP to achieve improvements in *mAP*. For instance, Yolo V4 only managed to obtain 67.9% *mAP* when using the SPP layer on its own, but when we combined the CSP and SPP layers, Yolo V4 CSP SPP achieved 78.84% *mAP*. When the SPP layer is considered, the *mAP* for the Yolo V3 model increases to 59.76% from its initial value of 58.86%. In addition, the accuracy of the Yolo V3 model improved to 59.91% after the CSP and SPP layers were integrated into it.

Yolo loss function is based on Equation (7) [40].

$$
\begin{aligned}
\lambda_{coord} \sum_{i=0}^{s^2} \sum_{j=0}^{B} \mathbb{1}_{ij}^{obj} &\left[ (x_i - \hat{x}_i)^2 + (y - \hat{y}_i)^2 \right] \\
&+ \lambda_{coord} \sum_{i=0}^{s^2} \sum_{j=0}^{B} \mathbb{1}_{ij}^{obj} \left[ \left( \sqrt{w_i} - \sqrt{\hat{w}_i} \right)^2 + \left( \sqrt{h_i} - \sqrt{\hat{h}_i} \right)^2 \right] \\
&+ \sum_{i=0}^{s^2} \sum_{j=0}^{B} \mathbb{1}_{ij}^{obj} \left( C_i - \hat{C}_i \right)^2 + \lambda_{noobj} \sum_{i=0}^{s^2} \sum_{j=0}^{B} \mathbb{1}_{ij}^{noobj} \left( C_i - \hat{C}_i \right)^2 \\
&+ \sum_{i=0}^{s^2} \mathbb{1}_i^{obj} \sum_{c\epsilon classes} (p_i \copyright \hat{p}_i(c))^2
\end{aligned}
\tag{7}
$$

where $\mathbb{1}_{ij}^{obj}$ denotes if the object appears in cell *i*, and $\mathbb{1}_{ij}^{obj}$ denotes that the $j^{th}$ bounding box predictor in cell *i* is responsible for the prediction. Next, $(\hat{x}, \hat{y}, \hat{w}, \hat{h}, \hat{c}, \hat{p})$ are used to express the anticipated bounding box's center coordinates, width, height, confidence, and category probability. Furthermore, our works set the *λcoord* to 0.5, demonstrating that the width and height errors are less useful in the computation. To reduce the impact of numerous grids, a loss value that is empty of objects, *λnoobj* = 0.5, is utilized.

The average mean average precision (*mAP*) is the integral over the precision *p(0)* and is described in Equation (8).

$$
mAP = \int_0^1 p(0) do
\tag{8}
$$

Where *p(0)* is the precision of the object detection. *IoU* calculates the overlap ratio between the boundary box of the prediction (*pred*) ground-truth (*gt*) and is shown in Equation (9). Precision and recall are represented by [41] in Equations (10) and (11).

$$
IoU = \frac{Area_{pred} \cap Area_{gt}}{Area_{pred} \cup Area_{gt}}
\tag{9}
$$

$$
Precision = \frac{TP}{TP + FP} = TP/N
\tag{10}
$$

$$
Recall = \frac{TP}{TP + FN}
\tag{11}
$$

where *TP* denotes true positives, *FP* denotes false positives, *FN* denotes false negatives, and *N* is the total number of objects recovered (including true positives and false positives). Another evaluation index, *F1* [42], is shown in Equation (12).

$$
F1 = \frac{2 \times Precision \times Recall}{Precision + Recall}
\tag{12}
$$

## 4. Results and Discussions

Table 2 shows the testing accuracy for all classes (bad, good, and none) of the MMD and FMD datasets. Based on the testing result, Yolo V4 CSP SPP scored the maximum *mAP* of 99.26% compared to other models in the experiment. Class ID 1 (good) achieved the highest average accuracy, around 96.47%, followed by Class ID 0 (bad) and Class ID 2 (none), with 64.92% and 58.32%, respectively. The second-highest model in the experiment is Yolo V4. This model exhibits *mAP* 74.26% and *IoU* 71.04%. Yolo V5 achieves the minimum *mAP* in the testing experiment, and it only achieved a *mAP* of 65.3%. From Table 2's experiments, we can deduce that adding CSP and SPP layers improves the accuracy of the test performance for all models.

**Table 2.** Testing accuracy performance for all models with FMD and MMD datasets.

| Model | Class ID | *AP* | Precision | Recall | F1-Score | *IoU* (%) | *mAP*@0.50 (%) |
|---|---|---|---|---|---|---|---|
| Yolo V3 | 0 | 55.6 | 0.84 | 0.86 | 0.85 | 66.32 | 67.11 |
|  | 1 | 95.65 |  |  |  |  |  |
|  | 2 | 50.08 |  |  |  |  |  |
| Yolo V3 CSP SPP | 0 | 60.76 | 0.82 | 0.89 | 0.85 | 63.67 | 69.41 |
|  | 1 | 95.97 |  |  |  |  |  |
|  | 2 | 51.51 |  |  |  |  |  |
| Yolo V3 SPP | 0 | 58.85 | 0.82 | 0.86 | 0.84 | 62.91 | 66.27 |
|  | 1 | 95.45 |  |  |  |  |  |
|  | 2 | 44.5 |  |  |  |  |  |
| Yolo V4 | 0 | 64.92 | 0.81 | 0.99 | 0.89 | 71.04 | 74.26 |
|  | 1 | 99.53 |  |  |  |  |  |
|  | 2 | 58.32 |  |  |  |  |  |
| Yolo V4 CSP SPP | 0 | 99.52 | 0.97 | 0.99 | 0.98 | 86.54 | 99.26 |
|  | 1 | 99.51 |  |  |  |  |  |
|  | 2 | 98.76 |  |  |  |  |  |
| Yolo V5 | 0 | 0.48 | 0.615 | 0.837 | 0.7 | 0.54 | 65.3 |
|  | 1 | 95.8 |  |  |  |  |  |
|  | 2 | 52.2 |  |  |  |  |  |

Figure 7 describes the results of recognizing the MMD and FMD datasets using Yolo V4 CSP SPP. Our proposed model detects objects in the image accurately. Yolo V4 CSP SPP can distinguish each class (bad, good, and none) by one object or multiple objects in one image. The Yolo V4 CSP SPP model exceeds the competition in medical masked face identification, which resulted in the introduction of the suggested model's effectiveness in medical masked face detection because of its superior performance.

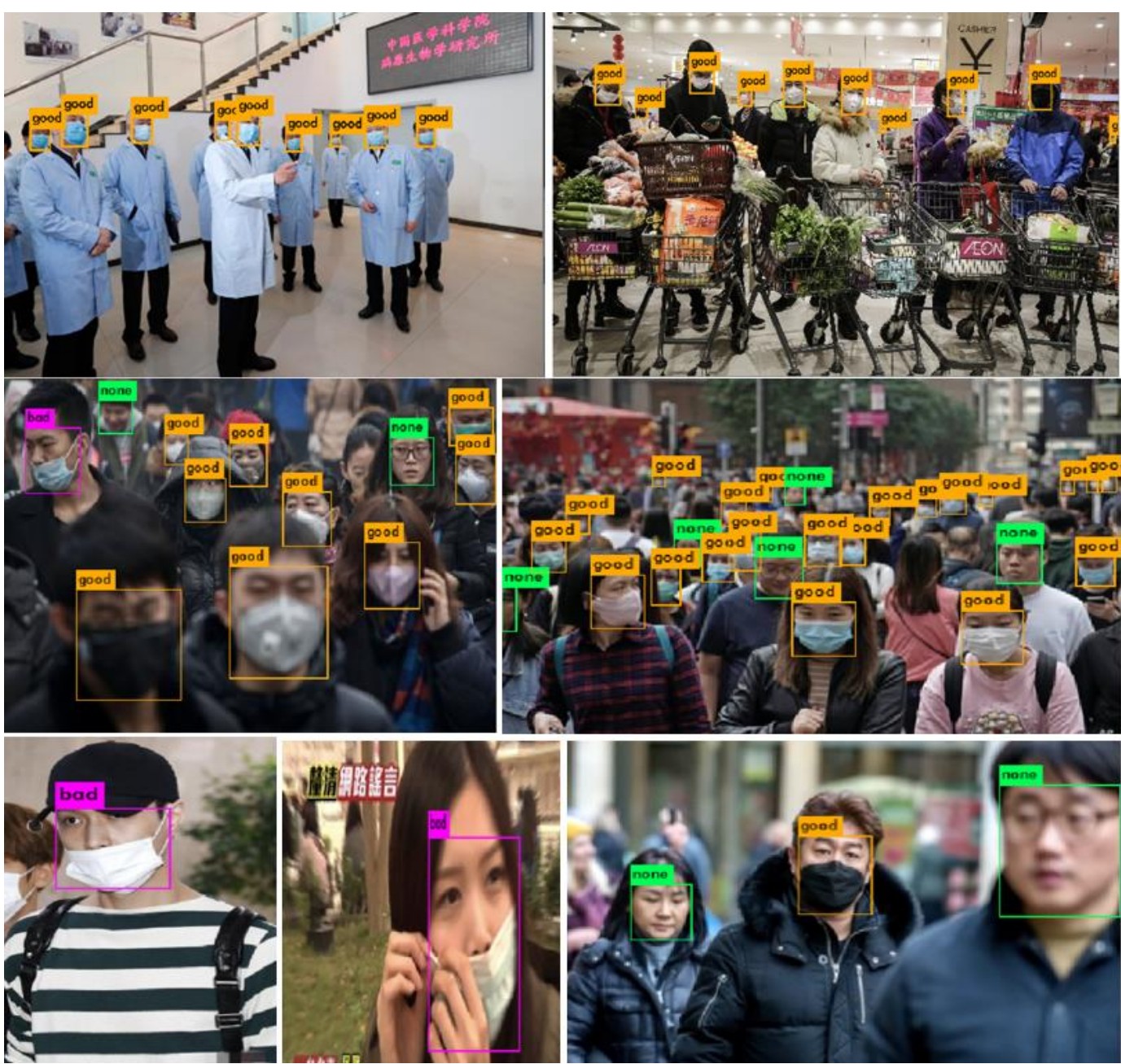

**Figure 7.** Recognition results using Yolo V4 CSP SPP. Source of the human faces: The FMD and MMD datasets.

CSP networks are designed to associate problems in network optimization with gradient information overload, which will allow for a large reduction in complexity while maintaining accuracy. In addition, pyramid pooling is not affected by the deformation generated by the object. SPP-net should be able to improve the performance of CNN-based image classification algorithms in general because of these advantages. Because our model can take advantage of CSP and SPP, all other models tested are able to produce the highest accuracy results when CSP and SPP are combined.

Failure detection is described in Figure 8. Sometimes, our model cannot detect bad classes and no classes because they are like each other. The bad class describes someone wearing the wrong mask. Usually, people wear masks but still show their noses. Because of this, our model shows error detection or double detection, as shown in Figure 8.

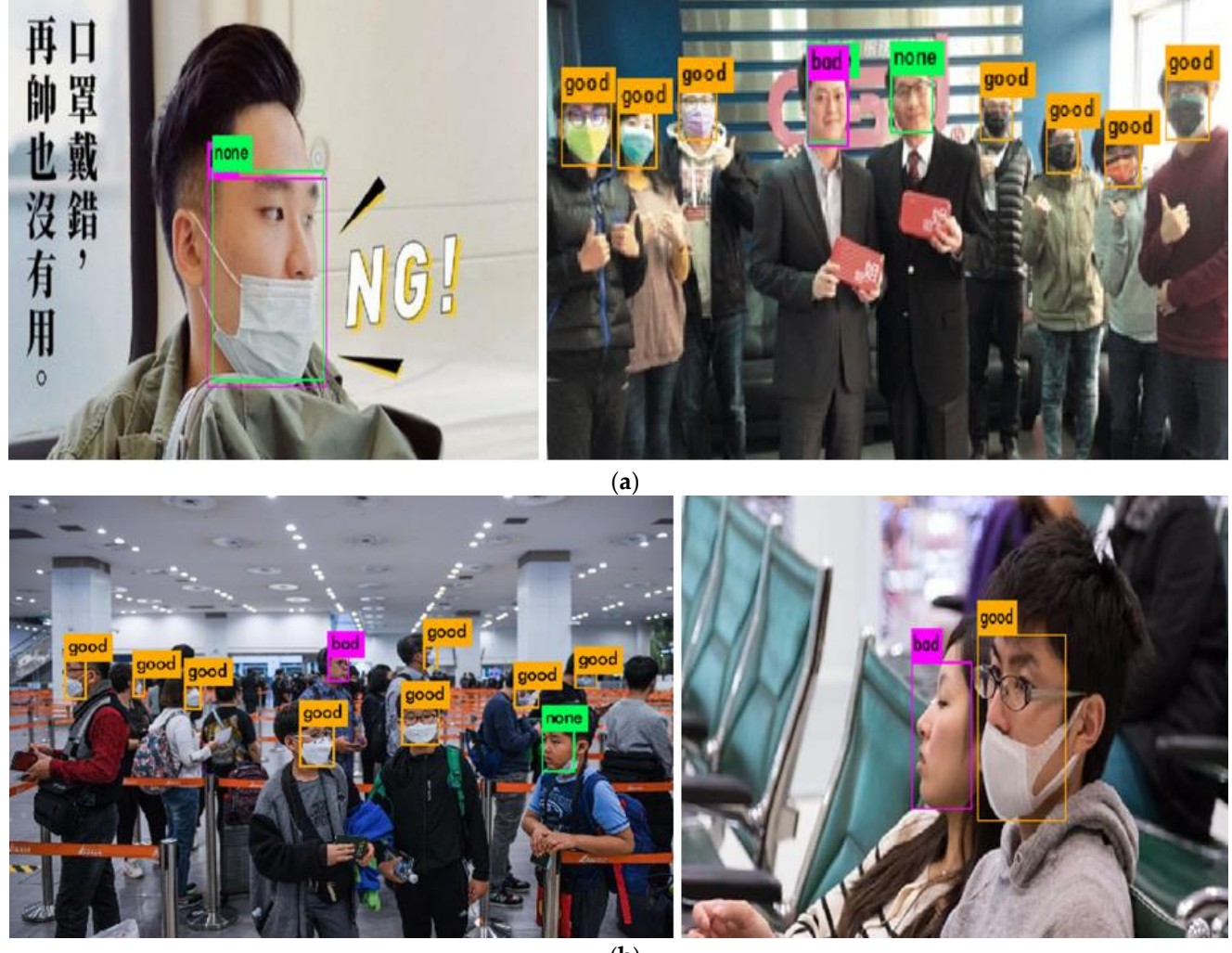

**Figure 8.** Failure Detection sample. (**a**) Face Mask Dataset (FMD) and (**b**) Medical Mask Dataset (MMD). Source of the human faces: The FMD and MMD datasets.

Billion floating-point operations (BFLOPS), workspace size, and layers for each CNN model were compared in the tests, as depicted in Figure 9. It produces a total of 65.304 BFLOPS and allocates an extra workspace size of 52.43 MB as well as loading 107 layers from a weights file. The Yolo V3 CSP SPP, in addition, loads 123 layers and requires a workspace size of 52.43 bytes, with an overall performance of 67.809 bytes per second per BFLOPS (bits per second squared). As the next step, the Yolo V4 CSP SPP and Yolo V4 load 162 and 176 layers, respectively, and provide a workspace size of 52.43 megabytes (MB) with a total of 59.569 and 75.142 BFLOPS. A record-breaking 213 layers are loaded into Yolo V5, which also boasts a performance of 109 BFLOPS and a workspace size of only 40.8 MB.

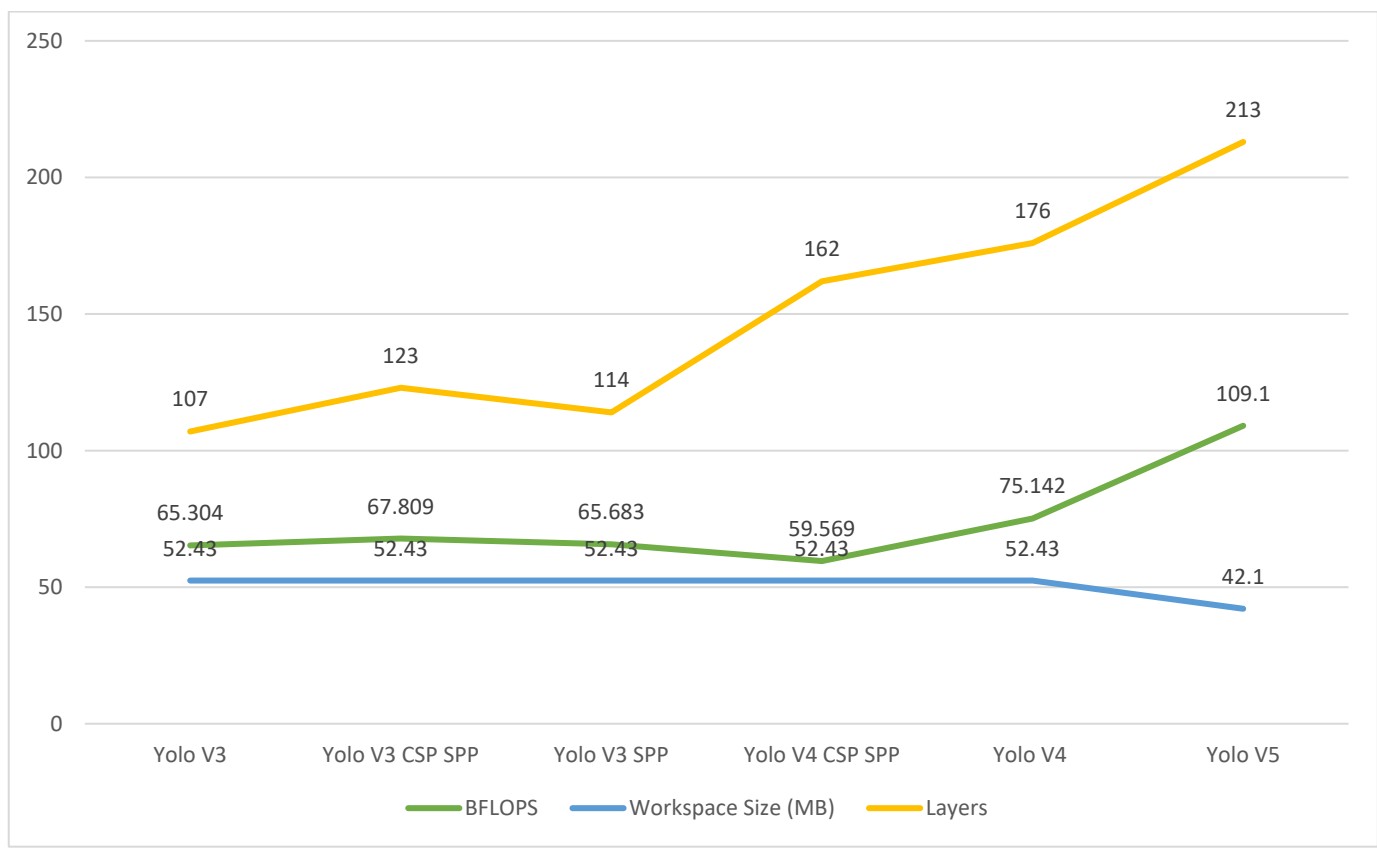

**Figure 9.** Comparison of BFLOPS, workspace size, and layers.

The previous research comparison explains in Table 3. Our proposed method outperforms other models in terms of accuracy with an accuracy of 99.26% with the FMD and MMD datasets. We increased the accuracy of previous studies with Yolo V2, with Resnet showing only 81% accuracy. Research in [17] proposed a hybrid method to perform classification only and reached 99.64% with the FMD dataset. When compared to prior research, the suggested technique achieves the highest accuracy in both classification and detection. Most of the related work focuses solely on the classification of masked faces. The classification and detection of medical masks are the primary objectives of our project.

The experimental solution tracks people with or without masks in real-time scenarios. If there is a violation at the scene or in a public area, our proposed algorithm can be paired with an alert to send out a warning. These analytics, which can be employed in a wide variety of public spaces, such as office buildings and airport terminals and gates, can be enabled by combining our suggested algorithm with the existing embedded camera infrastructure.

**Table 3.** Previous research comparison.

| Reference | Dataset | Methodology | Classification | Detection | Result *AP* (%) |
|---|---|---|---|---|---|
| (Ejaz et al., 2019) [13] | Our Database of Faces (ORL) | PCA | Yes | No | 70% |
| (Loey et al., 2021a) [17] | Face Mask Dataset (FMD) | Hybrid | Yes | No | 99.64% |
| (Ge et al., 2017) [16] | A Dataset of Masked Faces (MAFA) | LLE-CNNs | Yes | Yes | 76.4% |

| | | | | | |
|---|---|---|---|---|---|
| (Loey et al., 2021b) [3] | Face Mask Dataset (FMD) and Medical Mask Dataset (MMD) | Yolo V2 with Resnet | Yes | Yes | 81% |
| Proposed Method | Face Mask Dataset (FMD) and Medical Mask Dataset (MMD) | Yolo V4 CSP SPP | Yes | Yes | 99.26% |

The ablation studies are shown in Table 4. Moreover, our experiment used the COVID Face Mask Detection Dataset from Kaggle (https://www.kaggle.com/datasets/prithwirajmitra/covid-face-mask-detection-dataset, accessed on 13 January 2022). This dataset consists of two classes, "Mask" and "No Mask". We define "Mask" as class "Good" and "No Mask" to be class "None". In the experiment, we tested 50 images for each the "Mask" and "No Mask" classes with the proposed method, and a total of 100 images were tested. Based on Table 4, we can conclude that Yolo V4 CSP achieved the highest average accuracy of 97.0% for the "Good" class and 83.0% for the "None" class. Compare this with Yolo V4 which only achieved 92.0% average accuracy for the "Good" class and 82.0% for the "None" class. In addition, CSP can improve the performance of all models in experiments with the COVID Face Mask Detection Dataset. The proposed method is robust with other datasets.

**Table 4.** The ablation study with COVID Face Mask Detection Dataset.

| Model | Class | | | |
|---|---|---|---|---|
| | None | | Good | |
| | Acc (%) | Time (ms) | Acc (%) | Time (ms) |
| YoloV3 | 50.1 | 16.4 | 71.2 | 16.7 |
| YoloV3 CSP SPP | 61.4 | 18.1 | 76.1 | 18.4 |
| Yolo V3 SPP | 55.7 | 17.4 | 66.0 | 17.6 |
| Yolo V4 | 82.0 | 19.03 | 92.0 | 19.01 |
| Yolo V4 CSP | 83.0 | 19.07 | 97.0 | 19.08 |
| Yolo V5 | 72.7 | 11.13 | 85.7 | 11.02 |

Figure 10 exhibits the recognition result of COVID Face Mask Detection Dataset with Yolo V4 CSP. The proposed method Yolo V4 CSP can detect the "Good" and "None" classes correctly, as shown in Figure 9a,b.

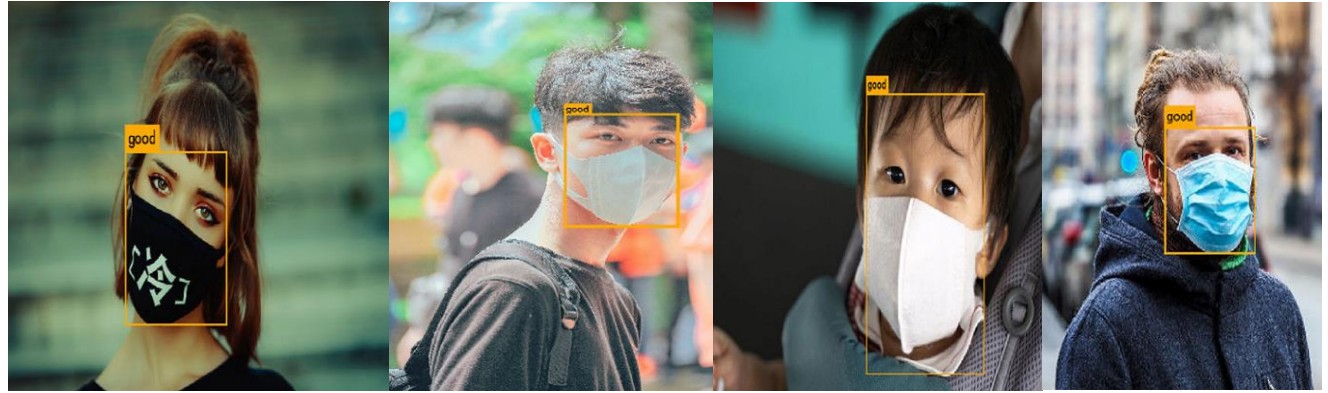

(**a**)

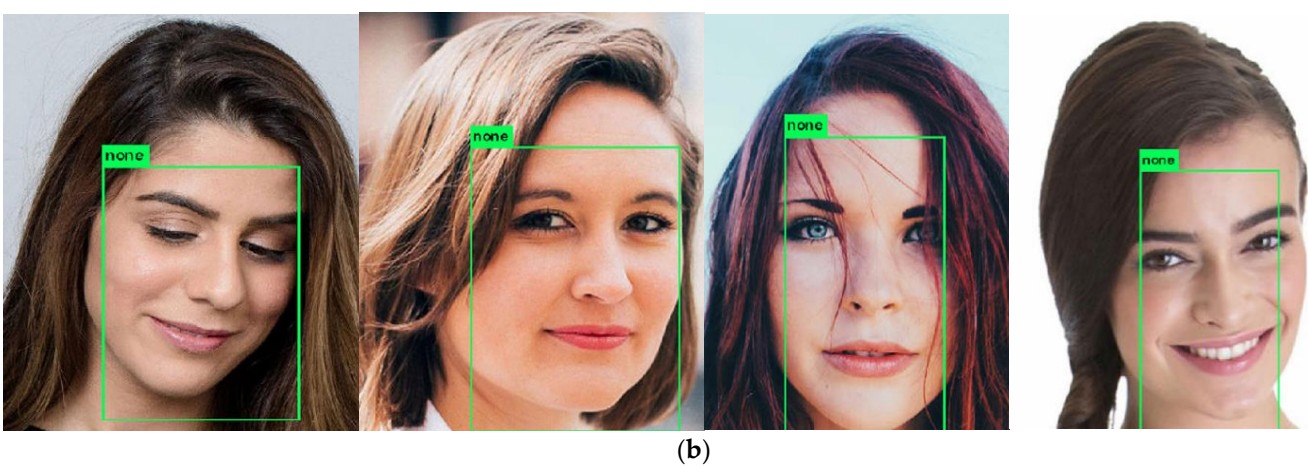

(**b**)

**Figure 10.** Recognition results of COVID Face Mask Detection Dataset with Yolo V4 CSP. (**a**) Class Good; (**b**) Class None. Source of the human faces: The COVID Face Mask Detection datasets.

## 5. Conclusions

The investigations of CNN-based object identification algorithms, specifically Yolo V3, Yolo V3 SPP, Yolo V3 CSP SPP, Yolo V4, Yolo V4 CSP SPP, and Yolo V5, are presented and analyzed in this paper. In each algorithm, we analyze and describe in detail the benefits that CSP and SPP bring to our experiments. According to the results of the experiments, Yolo V4 CSP SPP offers the highest possible level of precision. According to the findings of the experiments, both the CSP and SPP layers contribute to an increase in the accuracy of the CNN models' test performance. The proposed model takes full advantage of the benefits offered by both CSP and SPP.

To prevent the transfer of COVID-19 from one person to another, we provide in this study a unique model for medical masked face recognition that is focused on the medical mask object. The Yolo V4 SPP model was utilized to achieve high-performance outcomes in the field of medical mask detection. In comparison to the prior research studies using FMD and MMD datasets, the suggested model increases the detection performance of the previous research study from 81% to 99.26%. The experiments demonstrated that the Yolo V4 CSP SPP model scheme that we have suggested is an effective model for detecting medical face masks. Moreover, future research study will investigate the detection of a type of masked face in images and videos using deep learning models. Future study also will explore the explainable artificial intelligence (XAI) for medical mask detection.

**Author Contributions:** Conceptualization, C.D. and R.-C.C.; methodology, R.-C.C. and C.D.; software, C.D.; validation, C.D.; formal analysis, C.D.; investigation, R.-C.C.; resources, C.D.; data curation, C.D.; writing—original draft preparation, C.D. and R.-C.C.; supervision, R.-C.C.; project administration, C.D.; funding acquisition, R.-C.C. All authors have read and agreed to the published version of the manuscript.

**Funding:** This paper is supported by the Ministry of Science and Technology, Taiwan (MOST-111-2221-E-324-020).

**Institutional Review Board Statement:** Ethical review and approval were waived for this study, due to the reason that we use the public and free datasets from Kaggle, the FMD, MMD, and COVID Face Mask Detection datasets. The figures with human faces are from the public datasets.

**Informed Consent Statement:** Informed consent was waived for this study due to the reason that we use the public and free datasets from Kaggle, FMD, MMD, and COVID Face Mask Detection dataset and because the figures with human faces are from the public datasets.

**Data Availability Statement:** FMD Dataset (https://www.kaggle.com/datasets/andrewmvd/face-mask-detection, accessed on 13 January 2022), MMD Dataset (https://www.kaggle.com/da-tasets/shreyashwaghe/medical-mask-dataset, accessed on 24 January 2022), FMD+MMD Dataset

(https://drive.google.com/drive/folders/19uyNCP93wmBpmmVqi3ofFVAWLdLTdnMe?usp=sharing, accessed on 2 June 2022), COVID Face Mask Detection Dataset (https://www.kaggle.com/datasets/prithwirajmitra/covid-face-mask-detection-dataset, accessed on 24 August 2022).

**Acknowledgments:** The authors would like to thank all colleagues from Chaoyang Technology University and Satya Wacana Christian University, Indonesia and all involved in this research.

**Conflicts of Interest:** The authors declare no conflict of interest.

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
