# Peer review of "Automatic Medical Face Mask Detection Based on Cross-Stage Partial Network to Combat COVID-19"

_2504-2289, doi:10.3390/bdcc6040106_

Round 1

Reviewer 1 Report

The paper has an interesting topic and I would like the authors to do the following modifications and corrections to make it completed.

1. The authors should adjust the font size of some figures to fit the font of the full paper's text.

2. The authors should add descriptions of the contribution and innovation of the paper.

3. The authors should add how to apply relevant algorithms and research results to the real world.

Reviewer 2 Report

This paper focused on popular covid topic and proposed the deep learning based method to automatic detection the mask wearing. Introduction reviewed and summarized sufficient corresponding works and also covered the specific methods in related work. Here are some humble suggestions:

1. In Section 3.1 line 195,  I assume b denotes bounding box parameters or Yolo inputs, but there is no explaination in paper.

2. Figure 2 aims describing system structure, but the middle part of figure confused reader, what is relationship among the three modules?

3. Figure 5 lack the axis information, same as Figure 6

4. For training and testing, could you please add some content about how do you split the dataset? For example, if the test samples also appeared in training dataset? Or the size of training and test dataset?

5. Suppose you use all MMD and FMD dataset for training, then you will have an imbalanced dataset, 500 samples for none and bad classes each, but more than 4000 as good. And your training results also reflect this imbalanced situation because the good class has much higher AP than others. However, in the test results, especially YoloV4 CSP SPP case, the detection results seems be balanced back. Is there any discussions about this?

6. Overall, the test results is much better than training results which is not usual in deep learning. Is there something need be discussed? 

Round 2

Reviewer 2 Report

Thanks for your patient response. Most of responds are reasonable and clear.  

For your answer 4, the main point is you need describe the dataset split percentage like in answer 6. May add a few sentences about it.

For answer 5 and 6, your points are very clear. A few humble suggestions, maybe you can illustrate your data augmentation stage reduced data imbalance or not? Maybe a little discussion about this imbalance issue. Also if the test data which is from different env from training dataset is evaluated the trained model, the performance description may be more sufficient.

Round 3

Reviewer 2 Report

The added test results demonstrate the capacity of proposed model, The description became more completed and sufficient. Thanks for your hardworking.

1. It would be better that include a few images from the new dataset used in test. Also the result images with the bounding box labels.

2. The last minor suggestions: the added paragraph may use too may subjective word like "we"/"our".May need a minor edit about it.
